# Elevated Temperature Tensile Behavior of a Nb-Mo Microalloyed Medium Mn Alloy under Quasistatic Loads

**Wenlong Wu [1,2], Minghui Cai [1,3,*], Zeyu Zhang [1,2], Weigong Tian [1,2] and Haijun Pan [1,2]**

[1] School of Materials Science and Engineering, Northeastern University, Shenyang 110819, China; 20192490@stu.neu.edu.cn (W.W.); 20193028@stu.neu.edu.cn (Z.Z.); 20193071@stu.neu.edu.cn (W.T.); phjuning@163.com (H.P.)
[2] Key Laboratory of Lightweight Structural Materials, Northeastern University, Shenyang 110819, China
[3] State Key Laboratory of Rolling and Automation, Northeastern University, Shenyang 110819, China
[*] Correspondence: caimh@smm.neu.edu.cn; Tel.: +86-24-8367-2700; Fax: +86-24-8367-2600

**Abstract:** The elevated temperature tensile behavior of a Nb-Mo microalloyed medium steel was investigated over the −50 to 150 °C temperature range. The ultimate tensile strength was significantly reduced with increasing deformation temperature, but both YS (yield strength) and EI (total elongation) values changed slightly. The best product of UTS (ultimate tensile strength) and EI (~59.5 GPa·%) can be achieved at the deformation temperature of 50 °C, implying an excellent combination of strength and ductility. Furthermore, the change in strain hardening rate as a function of deformation temperature was further explained by the following two aspects: the dependence of mechanical stability of retained austenite on deformation temperature as well as the dependence of deformation mechanism on deformation temperature. Theoretical models and experimental observations demonstrate that the dominant deformation mechanism of the present medium Mn steel changed from the single transformation-induced plasticity (TRIP) effect at −50 to 50 °C to the multiple TRIP + TWIP (twinning-induced plasticity) effect at 50–150 °C.

**Keywords:** medium Mn steel; austenite stability; deformation temperature; transformation-induced plasticity (TRIP); twinning-induced plasticity (TWIP)





## 1. Introduction

Medium Mn steels (4–12 wt.%) as a candidate in the advanced high-strength steel family have gained widespread attention due to having a high strength level of ≥1000 MPa while maintaining excellent elongation of ≥30% [1–5]. The excellent overall tensile properties of medium Mn steels largely rely on the stability of austenite, which controls the activation of transformation-induced plasticity (TRIP) or twinning-induced plasticity (TWIP). Therefore, lots of efforts have been conducted to tailor the stability of retained austenite (RA) [6].

The stability of retained austenite is influenced by many factors, such as chemical compositions [7,8], grain size [9,10] and morphology [11,12]. Besides the above factors, deformation temperature is another key parameter affecting the stability of retained austenite and thus its transformation behavior [6,12]. This is partly because the transformation behavior of retained austenite depends on the stacking fault energy (SFE) of steel, which is usually a function of deformation temperature. Moreover, the difference in the chemical Gibbs free energy between austenite and martensite increases as the temperature decreases, which eventually affects the stability and deformation mechanism of retained austenite due to the additional driving force, such as applied stress [6]. Luo et al. reported that the ultimate tensile strength (UTS) of cold-rolled 7 wt.% Mn steel decreased with increasing tensile temperature owing to the decreasing volume fraction of transformed austenite [12]. Sugimoto et al. reported that deformation temperature significantly affected the mechanical properties of 0.2C-1.5Si-5Mn steel (wt.%) by affecting the morphology of strain-induced

martensite as well as the forest dislocation hardening behavior [13]. Tanino et al. pointed out that the fracture mechanism of 5 wt.% Mn steel was characterized by intergranular fracture at −196 °C, while cleavage fracture happened at 25 °C [14]. Kozlowska et al. [15] demonstrated that an increase in deformation temperature caused the reduced intensity of the TRIP effect due to the relatively high mechanical stability of retained austenite. Zhang et al. [16] stated that an optimal TRIP effect occurs in the temperature range of 25–50 °C for the Fe-0.12C-2Al-0.05Si steel; with increasing deformation temperature to 100 °C, both tensile strength and ductility drop. The dependence of austenite stability and deformation mechanism on the temperature in warm-rolled medium Mn steels has not yet been fully understood.

In the present study, we investigated the elevated temperature tensile behavior of a Nb-Mo microalloyed medium steel under quasistatic loads. Further, the influence of deformation temperature on tensile properties and strain hardening behavior was analyzed. Based on the detailed microstructural examinations and transformation thermodynamics models, the dependence of austenite stability and deformation mechanism on deformation temperature was also discussed.

## 2. Materials and Methods

A medium Mn steel with a chemical composition of Fe-0.19C-5.6Mn-1.2Al-0.05Nb-0.22Mo (wt.%) was employed for investigation. The multiple additions of Nb-Mo were expected to improve the yield strength significantly by various strengthening mechanisms, including nanoprecipitation and grain refinement. A round ingot was smelted in a 50 kg vacuum induction furnace, heated to 1200 °C for 1 h and then forged into rectangular slabs. Small billets of 100 mm × 100 mm × 40 mm were cut from the as-received slabs and then were reheated to 800 °C for 1 h for austenitization. Multipass warm rolling was performed using a four-high rolling machine, in the temperature range from 400 to 700 °C with approximately 95% total reduction. The final sheet thickness was about 1.8 to 2.3 mm.

These specimens were subjected to intercritical annealing at 650 °C for 1 h and then air-cooled to room temperature. Tensile specimens were prepared parallel to the rolling direction with a gauge section of 25 mm × 6 mm, according to the ASTM E8/E8M standard. Uniaxial tensile tests were carried out at −50, 0, 25, 50, 100 and 150 °C with a constant strain rate of $5 \times 10^{-3}$ s$^{-1}$. Tensile testing was conducted in a closed high–low temperature chamber. All samples were held in the chamber for 15 min prior to tensile testing for the temperature homogeneity.

Microstructural observations were performed using a field-emission scanning electron microscope (FE-SEM, Supra, SSX-550, Shimadzu, Kyoto, Japan) and a field-emission transmission electron microscope (FE-TEM, FEI, Tecnai G2 F20, Hillsboro, OR, USA, operated at 200 kV) combined with energy-dispersive X-ray spectroscopy (EDXS, INCA Energy, Oxford, UK). SEM specimens were etched with 15% NaHSO$_3$ water solution after standard mechanical polishing. TEM specimens were electropolished in a solution of 95% CH$_3$COOH and 5% HClO$_4$ at 18 °C using a Twin-Jet polisher (Struers, Tenupol-5, Copenhagen, Denmark).

The volume fraction of retained austenite was determined by X-ray diffraction (XRD, Rigaku, D/Max2250/PC, Tokyo, Japan) using a Cu-K$\alpha$ radiation source ($\lambda$ = 1.5405 Å), based on a direct comparison of the integrated intensities of all diffraction peaks [17]. The scanning range, speed and step size were 40–100°, 2° min$^{-1}$ and 0.02°, respectively.

## 3. Results

Figure 1a shows the SEM microstructure of the warm-rolled medium Mn steels after intercritical annealing at 650 °C for 1 h. The optimal heat-treatment processing parameters have been detailed in our preliminary work [18,19]. The phases of the annealed samples are identified as ferrite ($\alpha$) and retained austenite ($\gamma$) grains, based on their morphological features, as arrowed in yellow. Furthermore, both phases had two types of lamellar and equiaxed morphologies, which is totally different from the cold-rolled fully equiaxed morphology or the hot-rolled fully lamellar morphology [1,4,7,8].

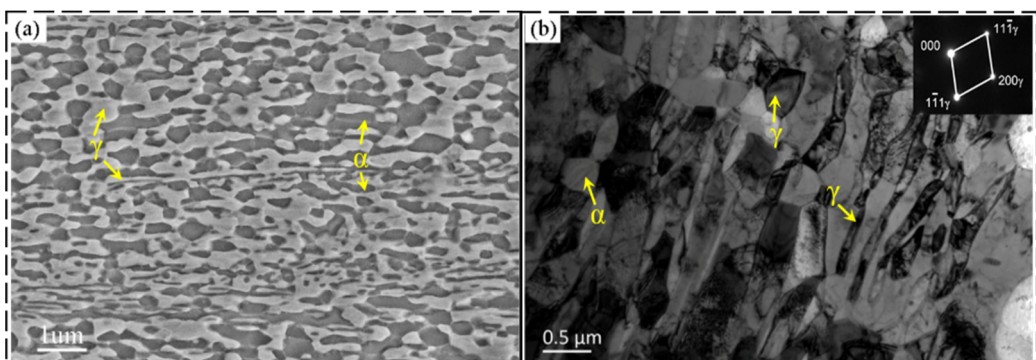

**Figure 1.** SEM (**a**) and TEM (**b**) micrographs of experimental steel after intercritical annealing at 650 °C for 1 h.

In order to observe the fine microstructure of ferrite and austenite phases, TEM was performed, and their representative morphologies are displayed in Figure 1b. For the intercritically annealed samples, the average grain size or width of both phases is less than 0.5 μm, and the dislocation density is relatively low. Furthermore, no annealing twins can be observed in these samples.

The tensile curves and the corresponding tensile properties of the annealed samples after tensile deformation at various temperatures are displayed in Figure 2. All the engineering stress–strain curves exhibit clear yielding plateaus, followed by strain hardening at the uniform plastic deformation stage. The presence of serrated features at the temperature between 50 and 100 °C indicates that dynamic strain aging and strong localized deformation happen. As the deformation temperature was approximately −50 °C, the sample exhibited the highest ultimate tensile strength (UTS) of ~1521 MPa and yield strength (YS) of ~1037 MPa as well as the lowest total elongation (EI) of ~36%. With the increase in the deformation temperature, the UTS value significantly reduced, but both YS and EI values changed slightly. Thus, an increase in deformation temperature leads to an obvious decrease in strain hardening rate. In metastable austenitic steels, it is generally known that there is a direct relationship between the strain hardening rate and the martensitic transformation rate [20]. The higher martensitic transformation rate corresponds to the higher strain hardening rate. In this case, the reduced strain hardening rate with increasing deformation temperature can be considered to be closely related to the mechanical stability of the austenite. The detailed reason will be discussed later. Furthermore, a careful comparison with tensile data shows that the best product of UTS and EI (~59.5 GPa·%) can be achieved at the deformation temperature of 50 °C, implying an excellent combination of strength and ductility.

To evaluate the mechanical stability retained austenite of the present medium Mn steels during tensile deformation at various temperatures, a series of X-ray diffraction observations have been conducted, and the results are shown in Figure 3a,b. The changes in the phase fraction of retained austenite before and after tensile testing as a function of deformation temperature are summarized in Figure 3c. It can be seen that the volume fraction of retained austenite almost remains constant (>35%) before tensile deformation in the temperature range from 0 to 150 °C, while it decreases to less than 30% at −50 °C. This may be the reason that the deep cryogenic treatment below the $M_s$ led to the further transformation of retained austenite to martensite. After tensile deformation, the fraction of retained austenite dramatically drops in the designed temperature range, implying the occurrence of retained austenite to martensitic transformation upon loading. However, the extent of martensitic transformation is associated with deformation behavior; i.e., the relatively lower fraction of austenite transforms to martensite at the relatively higher deformation behavior. As seen in Figure 3d, the rate of martensite transformation ($dV_{\alpha'}/d\varepsilon$) clearly demonstrates that the transformed martensite fraction was about 30% in the temperature ranged from −50 to 50 °C, and then it decreased quickly for the higher deformation

temperature. Thus, it can be inferred that an increase in deformation temperature decreases the transformation ratio of retained austenite to martensite.

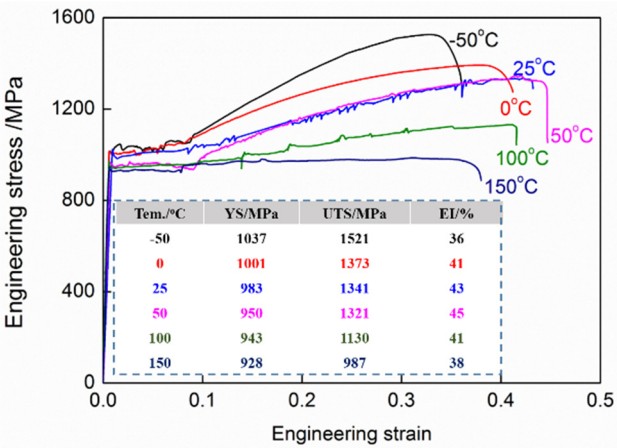

**Figure 2.** Engineering stress–strain curves and mechanical properties of tensile samples at −50, 0, 25, 50, 100 and 150 °C.

**Figure 3.** XRD patterns before (**a**) and after (**b**) the tensile deformation, volume fraction of austenite before and after the tensile deformation (**c**), transformed austenite and transformation ratio of austenite (**d**) of the annealed samples deformed at −50, 0, 25, 50, 100 and 150 °C.

## 4. Discussion

The above-mentioned experimental results indicate that the present warm-rolled medium Mn steels exhibit a strong temperature dependence, especially in their ultimate tensile strength and strain hardening. The elevated temperate tensile behavior is mainly attributed to the combined effects of the stability of retained austenite and the deformation mechanism. In this section, the dependence of austenite stability and deformation mechanism on deformation temperature is discussed in detail, based on the transformation thermodynamics models and TEM analyses.

### 4.1. The Dependence of Stability of Retained Austenite on Deformation Temperature

The mechanical stability of retained austenite is a key parameter to influence the mechanical properties of TRIP steels [21–27], and it can be estimated by the stress-assisted martensitic transformation thermodynamics model [21,22]:

$$\Delta G^{chem} + \Delta G^{mech} > E^{store} \tag{1}$$

where $\Delta G^{chem}$ is the chemical driving stimulus, $\Delta G^{mech}$ is the mechanical driving stimulus and $E^{store}$ is the stored energy of martensite. $\Delta G^{chem}$ can be calculated by Equation (2) [26,27]:

$$\Delta G^{chem} = x_c(16,277 - 8.08T - 17,660x_c - 10,066x_{Mn}) + x_{Mn}(-430 + 0.305T) \\ + x_{Mn}(1 - x_{Mn})(-6500 + 3.7T) + (1 - x_c - x_{Mn})\Delta G_{Fe}^{\gamma \to \alpha} \tag{2}$$

where $x_c$, $x_{Mn}$, $T$ and $\Delta G_{Fe}^{\gamma \to \alpha}$ are the mole fraction of C, the mole fraction of Mn, the temperature in Kelvin and the Gibbs free energy difference of the pure iron from austenite to ferrite, respectively [26]. If austenite transforms to martensite via single variant mode, the increase in elastic strain energy per unit volume ($\Delta E_v$) is given as follows [22]:

$$\Delta E_v = 1276.1(x/d)^2 + 562.6(x/d) \tag{3}$$

where $x$ (about 0.11 μm based on TEM) and $d$ (about 0.4 μm based on TEM) are the thickness of martensite plate and austenite grain size, respectively. The mechanical driving force can also be expressed as follows [21,22]:

$$\Delta G^{mech} = 0.86\sigma \tag{4}$$

where $\sigma$ is the external stress. A higher $\sigma$ is necessary for the TRIP effect at higher deformation temperature, which is mainly attributed to the decreased $\Delta G^{chem}$ at higher deformation temperature, as shown in Figure 4. Therefore, a higher deformation temperature can increase the mechanical stability of retained austenite. As a result, the present medium Mn steel exhibits a reduced strain hardening rate with increasing deformation temperature, thus causing the significantly reduced ultimate tensile strength.

### 4.2. The Dependence of Deformation Mechanism on Deformation Temperature

It is well known that the transformation behavior of retained austenite during deformation greatly relies on the stacking fault energy (SFE) [9]. For example, an SFE of 15–20 mJ/m$^2$ favors the formation of deformation twins, while an SFE below 15 mJ/m$^2$ favors the martensitic formation [27]. The SFE value can be expressed by the following equation [9]:

$$\text{SFE} = 2\rho\Delta G^{\gamma \to \varepsilon} + 2\sigma \tag{5}$$

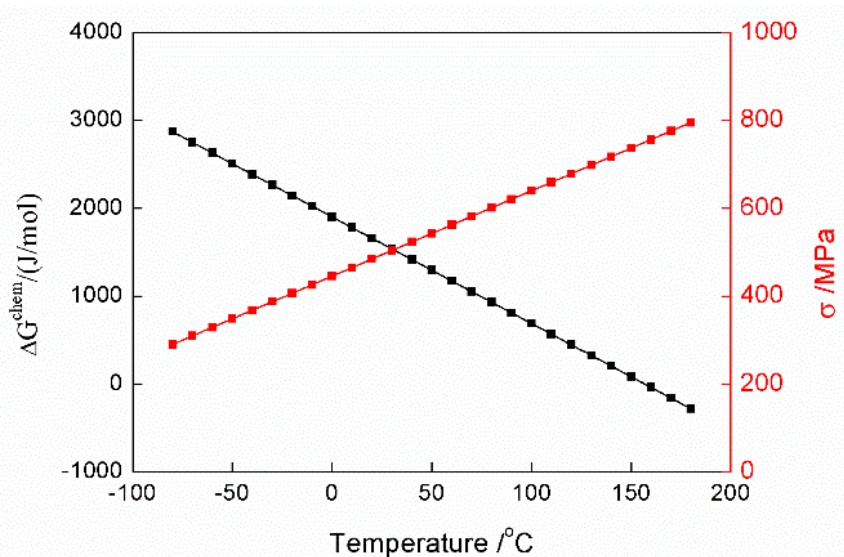

**Figure 4.** Chemical driving stimulus for martensite transformation ($\Delta G^{chem}$) and the external stress ($\sigma$) as a function of deformation temperature.

where $\rho$, $\Delta G^{\gamma \to \varepsilon}$ and $\sigma$ are the molar surface density along {111} planes, the molar Gibbs energy for $\gamma \to \varepsilon$ phase transformation, and the {111} interface energy between $\gamma$ and $\varepsilon$ (~9 mJ/m$^2$ [26]), respectively. $\rho$ can be expressed using the following equation [9]:

$$\rho = \frac{4}{\sqrt{3}} \frac{1}{a_\gamma^2 N} \tag{6}$$

where $N$ and $a_\gamma$ are the Avogadro constant and the average lattice parameter of austenite (~0.359 nm [9]), respectively. For the Fe-Mn-Al-C alloy system, the $\Delta G^{\gamma \to \varepsilon}$ can be calculated by the following equation [7,17]:

$$\Delta G^{\gamma \to \varepsilon} = \chi_{Fe} \Delta G_{Fe}^{\gamma \to \varepsilon} + \chi_{Mn} \Delta G_{Mn}^{\gamma \to \varepsilon} + \chi_C \Delta G_C^{\gamma \to \varepsilon} + \chi_{Al} \Delta G_{Al}^{\gamma \to \varepsilon} + \chi_{Fe} \chi_{Mn} \Delta \Omega_{FeMn}^{\gamma \to \varepsilon} + \chi_{Fe} \chi_{Al} \Delta \Omega_{FeAl}^{\gamma \to \varepsilon}$$
$$+ \chi_{Fe} \chi_C \Delta \Omega_{FeC}^{\gamma \to \varepsilon} + \chi_{Mn} \chi_C \Delta \Omega_{MnC}^{\gamma \to \varepsilon} + \Delta G_{Mn}^{\gamma \to \varepsilon} \tag{7}$$

where $\chi_M$ and $\Delta G_M^{\gamma \to \varepsilon}$ present the atomic fraction and molar Gibbs energy of pure element (Fe, Mn, Al, C), $\Delta \Omega_{Mm}^{\gamma \to \varepsilon}$ is the interaction energy parameter of the binary systems, and $\Delta G_{Mn}^{\gamma \to \varepsilon}$ is the molar Gibbs energy due to the magnetic state of the phase [9]. The necessary thermodynamic data of Equations (5)–(7) are listed in Table 1.

The calculated values of SFE as a function of deformation temperature are shown in Figure 5. The increase in deformation temperature from −50 to 150 °C leads to the increase in SFE from 12.4 to 17.5 mJ/m$^2$. Therefore, from the view of the calculated results, the transformation behavior of retained austenite may turn from the TRIP effect to the TRIP + TWIP effect with increasing deformation temperature.

In order to further confirm the transformation behavior of retained austenite, typical TEM morphologies of the annealed samples after tensile deformation at −50, 50 and 150 °C are shown in Figure 6. After tensile deformation, no deformation twins exist at −50 °C (Figure 6a), while many deformation twins can be observed at both 50 and 150 °C (Figure 6b,g). Moreover, the amount of deformation twins at 150 °C (Figure 6g) is relatively higher than that at 50 °C (Figure 6b). In addition, the dislocation density at −50 °C (Figure 6b) is much higher than that at 150 °C (Figure 6h), which may be related to the higher annihilation rate of dislocation at higher temperature [14–16].

**Table 1.** Thermodynamic parameters and equations to calculate $\Delta G_M^{\gamma \to \varepsilon}$ and $\Delta \Omega_{Mm}^{\gamma \to \varepsilon}$ [7].

| Parameter | Function |
|---|---|
| $\Delta G_{Fe}^{\gamma \to \varepsilon}$ | $-2243.38 + 4.309T$ (J/mol) |
| $\Delta G_{Mn}^{\gamma \to \varepsilon}$ | $-1000 + 1.123T$ (J/mol) |
| $\Delta G_{C}^{\gamma \to \varepsilon}$ | $-22,166$ (J/mol) |
| $\Delta G_{Al}^{\gamma \to \varepsilon}$ | $2800 + 5T$ (J/mol) |
| $\Delta \Omega_{FeMn}^{\gamma \to \varepsilon}$ | $2873 - 717(\chi_{Fe} - \chi_{Mn})$ (J/mol) |
| $\Delta \Omega_{FeAl}^{\gamma \to \varepsilon}$ | $3339$ (J/mol) |
| $\Delta \Omega_{FeC}^{\gamma \to \varepsilon}$ | $42,500$ (J/mol) |
| $\Delta \Omega_{MnC}^{\gamma \to \varepsilon}$ | $26,910$ (J/mol) |
| $\Delta G_{Mn}^{\gamma \to \varepsilon}$ | $\Delta G_{Mn}^{\varepsilon} - \Delta G_{Mn}^{\gamma}$ |
| $\Delta G_{Mn}^{\Theta}$ | $f\left(\frac{T}{T_N^{\Theta}}\right) RT ln\left(1 + \frac{\beta^{\Theta}}{\mu_B}\right)$, $\Theta = \gamma, \varepsilon$ |
| $f\left(\frac{T}{T_N^{\Theta}}\right)$ | $1 - \dfrac{\frac{79\tau^{-1}}{140p} + \frac{474}{497}\left[\frac{1}{P} - 1\right]\left[\frac{\tau^3}{6} + \frac{\tau^9}{135} + \frac{\tau^{15}}{600}\right]}{D}$ <br> When $\tau \le 1$ <br> $f\left(\frac{T}{T_N^{\Theta}}\right) = -\left[\dfrac{\frac{\tau^{-5}}{10} + \frac{\tau^{-15}}{315} + \frac{\tau^{-25}}{1500}}{D}\right]$ <br> When $\tau > 1$ <br> Where $\tau = \frac{T}{T_N^{\Theta}}$ $P = 0.28$, $D = 2.34$ |
| $\frac{\beta^{\gamma}}{\mu_B}$ | $0.7\chi_{Fe} + 0.62\chi_{Mn} - 0.64\chi_{Fe}\chi_{Mn} - 4\chi_C$ (J/mol) |
| $\frac{\beta^{\varepsilon}}{\mu_B}$ | $0.62\chi_{Mn} - 4\chi_C$ (J/mol) |
| $T_N^{\gamma}$ | $251.71 + 681\chi_{Fe} - 1575\chi_{Fe} - 1740\chi_{Fe}$ (K) |
| $T_N^{\varepsilon}$ | $580\chi_{Mn}$ (J/mol) |

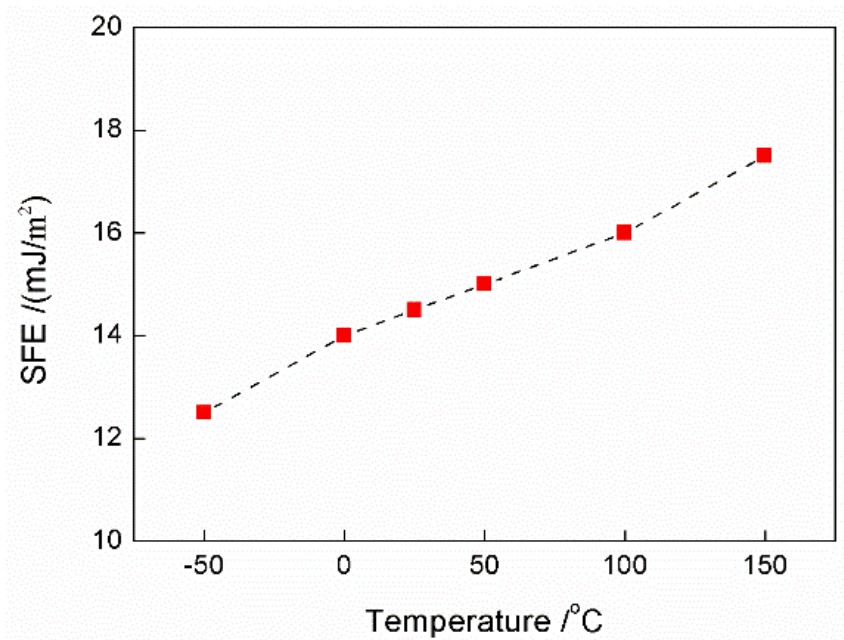

**Figure 5.** The values of SFE as a function of deformation temperature.

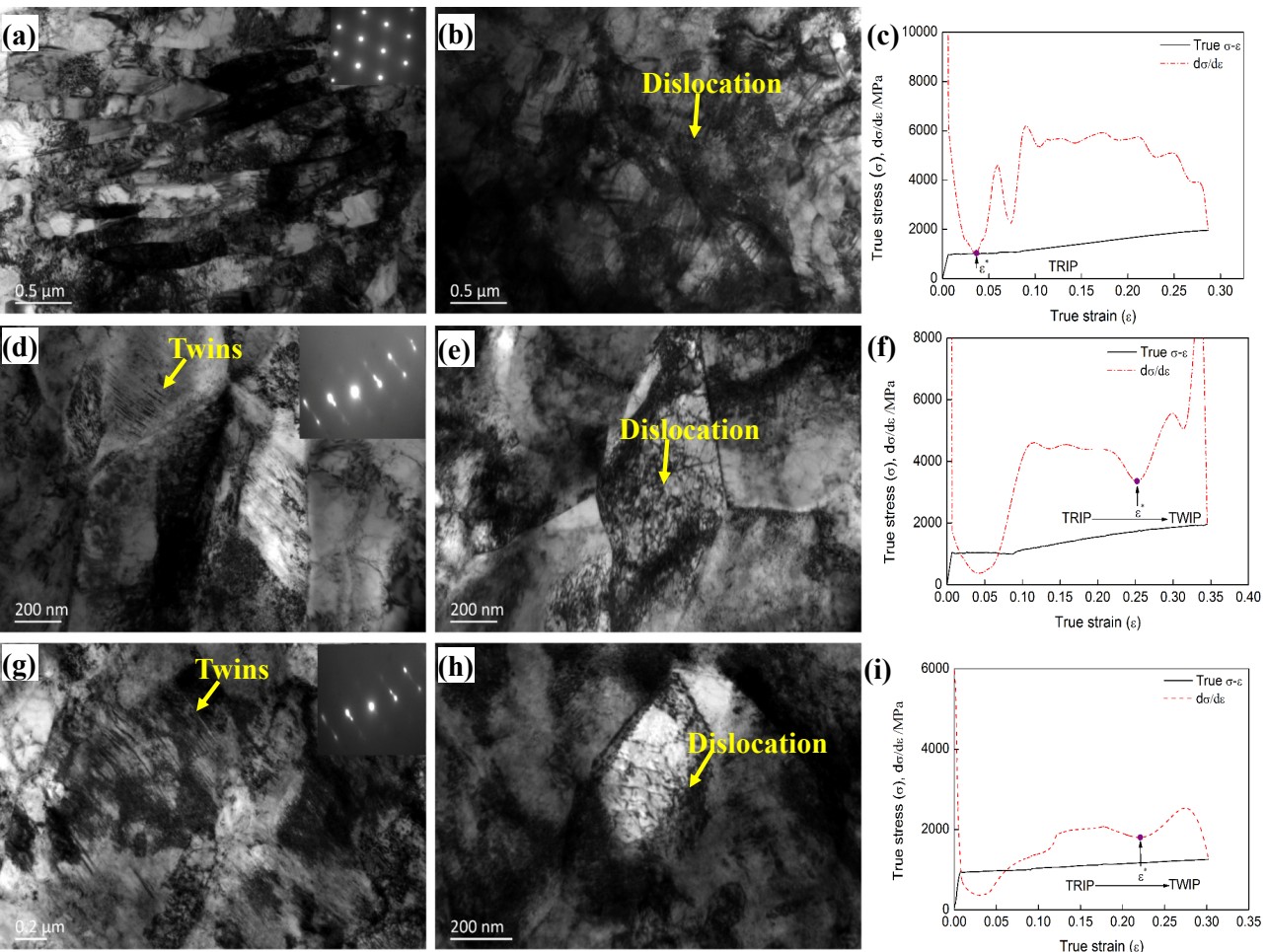

**Figure 6.** TEM morphologies after tensile failure and work hardening rate curves of samples deformed at −50 (**a–c**), 50 (**d–f**) and 150 °C (**g–i**).

Figure 6c,f,i also shows the corresponding true stress ($\sigma$) versus true strain ($\varepsilon$) curves together with the strain hardening rate ($\theta = \mathrm{d}\sigma/\mathrm{d}\varepsilon$) for three samples deformed at −50, 50 and 150 °C. All samples exhibit multiple-stage strain-hardening behavior features. Surprisingly, the sample deformed at −50 °C exhibits a decreased strain hardening rate with true strain at the later stage of deformation, whereas those samples deformed at both 50 and 150 °C show an increased strain hardening rate with true strain at the later stage of deformation. Such a distinct change in $\theta$–$\varepsilon$ curves from the −50 to 50 °C (or 150 °C) deformation temperature conditions reveals the importance of the combined effects of austenite stability and its volume fraction on the tensile behavior of the present warm-rolled medium Mn steel. In general, an increase in the UTS and strain hardening rate of medium Mn steels is associated with the increase in hard phase (martensite) and dislocation density upon loading [14]. Definitely, the TRIP effect promotes the martensitic transformation and increases the fraction of the hard phase (martensite). Based on the above TEM observation and multiple-stage strain-hardening behavior features of $\theta$−$\varepsilon$ curves, a continuous increase in $\theta$ value at the initial early stage of uniform plastic deformation implies the occurrence of TRIP, whereas a continuous increase in $\theta$ value at the later early stage of uniform plastic deformation implies the occurrence of TWIP. Thus, it can be noted that the dependence of the deformation mechanism on temperature is visible; i.e., both TRIP and TWIP effects occur at 50 and 150 °C, while only the TRIP effect happens at −50 °C. These experimental observations are in broad agreement with the above theoretically calculated results. Accordingly, it can be concluded that the relatively higher UTS and strain hardening rate in the samples deformed at −50 °C account for the higher dislocation density and TRIP effect,

while the samples deformed at 150 °C exhibit a lower UTS due to the lower dislocation density and the limited TRIP effect.

## 5. Conclusions

In this paper, the stability of retained austenite and the deformation behavior of a novel Nb-Mo added warm-rolled medium Mn steel at different deformation temperatures were investigated. The main conclusions are described as follows:

(1) The higher deformation temperature increases the stability of retained austenite for the lower chemical driving force for transformation from retained austenite to martensite. The increase in deformation temperature leads to a decrease in the fraction of transformed austenite from 30% to 9%.

(2) The ultimate tensile strength was significantly reduced with increasing deformation temperature, but both YS and EI values changed slightly. The best product of UTS and EI (~59.5 GPa·%) can be achieved at the deformation temperature of 50 °C, implying an excellent combination of strength and ductility.

(3) The increase in deformation temperature from −50 to 150 °C leads to the increase in the SFE value from 12.4 to 17.5 mJ/m$^2$. When the deformation was −50 °C, the deformation mechanism was mainly the TRIP effect, while the deformation mechanism was mainly governed by the TWIP effect at 150 °C.

**Author Contributions:** Conceptualization, M.C.; methodology, investigation, W.W., Z.Z. and W.T.; writing—W.W., H.P.; writing—review and editing, M.C.; supervision, M.C. All authors have read and agreed to the published version of the manuscript.

**Funding:** The research was funded by the National College Student Innovation and Entrepreneurship Training Program (No. 200103); Natural Science Foundation of China (No. 51671149); Fundamental Research Funding of the Central Universities, China (Nos. N2002002 and N180702012) and Key R&D and Promotion Special Project of Henan Province (No. 212102210444).

**Data Availability Statement:** All data, models and data post-processing approaches in this study are available from the corresponding author by request.

**Acknowledgments:** Special thanks are due to the instrumental/data analysis from Analytical and Testing Center, Northeastern University.

**Conflicts of Interest:** The authors declare no conflict of interest.

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
