# Peer review of "Elevated Temperature Tensile Behavior of a Nb-Mo Microalloyed Medium Mn Alloy under Quasistatic Loads"

_metals, doi:10.3390/met12030442_

Round 1
Reviewer 1 Report
The authors studied the influence of tensile temperature on mechanical properties of a Nb-Mo microalloyed medium steel was studied. The change in austenite stability and deformation mechanisms with tensile temperature was revealed by combing transmission electron microscopy (TEM) characterization and theoretical calculation.
The manuscript could be accepted after major revision.
The literature should be supported by published articles in metals.
The abstract and introduction should be improved.
What are the precautions that have been applied during testing under -50°C?
The mechanical and thermal properties of the used alloys should be included.
Tem images (Fig 6) are very poor.
Why did coarse recrystallized grains produce?
The stress-strain curves should be carefully discussed.
The discussions are very weak.
Why do UTS, YS and work hardening rate decrease with increasing the deformation temperature.
Author Response
Thank you for the reviewer's valuable suggestion to further improve the quality of this manuscript. A point-buy-point response has been provided in the attached file.

Reviewer 2 Report
The transformation of the retained austenite through TRIP and TWIN effects is known already many decades. On the other hand, the stability of austenite in the chosen alloy is described very precisely using all necessary experimental methods of materials science. Theoretically the stability of austenite is described on the base of thermodynamic potentials including their numerical values respecting the chemical composition of the studied steel and experimental conditions.
Generally
The manuscript clearly written and well-structured.
The half of the cited references is younger than 5 years, only 3 references are older than 10 years (original papers are referred). No self-citations appear.
All experimental conditions are described sufficiently so that they can be fully reproduced.
All figures are appropriate. True stress-strain curves with their differentiations in Fig. 6 could have been used more effectively in the Discussion.
The conclusions of the manuscript naturally follow from the presented results.
No ethical deficiencies were found.
Specific comments
Eq. (1) and surrounding text: thermodynamic potentials have the dimension of (specific) energy. They can only be called as “driving forces” in a figurative sense, formally “energy” and “force” are different physical quantities. Maybe “driving stimulus” could be considered.
Formal comments
Physical units should be written in Antique, separated by space from numerical value (including temperature values, e.g. –50 °C).
“Avogadro” is a family name, starting with capital letter.
Line 133: correct “deforamtion”.
Author Response
Thank you for your suggestion. Some errors have been updated as follows: 1) The driving force has been replaced by the driving stimulus in the revised version; 2) A space has been added between physical units and numerical value; 3) The letter of Avogadro starts with a capital letter. 4) The other errors have also been revised in the revised version.
Reviewer 3 Report
The manuscript reports the investigation of mechanical properties of Nb-Mo microalloyed medium steels. In general, it is well written, and the characterisation is properly carried out. However, the authors should improve the data discussion, modelling the observed behaviour and discussing the data on the light of existing literature. Actually, it can be considered a report of the study, not a reaserch paper.
Details about the experimental parameters used for the characterisation could be reported to garantee the reproducibility of the experiments.
The quality of Figures and in particular of Fig6 should be improved. Some errors of type are present.
The manuscript can be taken in consideration for publication after majior revision.
Author Response
1) Thank you for your valuable suggestion to improve the quality of the manuscript. According to the reviewer 1, most of the draft manuscript has been updated from the abstract to the conclusion. Please see the revised version.
2) In the experimental procedure, the following aspect was added in the revised version: Tensile testing was done in a closed high-low temperature chamber. Also, all samples were held in the chamber for 15 min prior to tensile testing for the temperature homogeneity.
3) All original figures have been provided in the attachment.

Round 2
Reviewer 1 Report
Accept in present form
Reviewer 3 Report
The authors extensive modified the manuscript. In the present version its quality improved.I support the publication of the manuscript.